# Consistent123: One Image to Highly Consistent 3D Asset Using Case-Aware Diffusion Priors

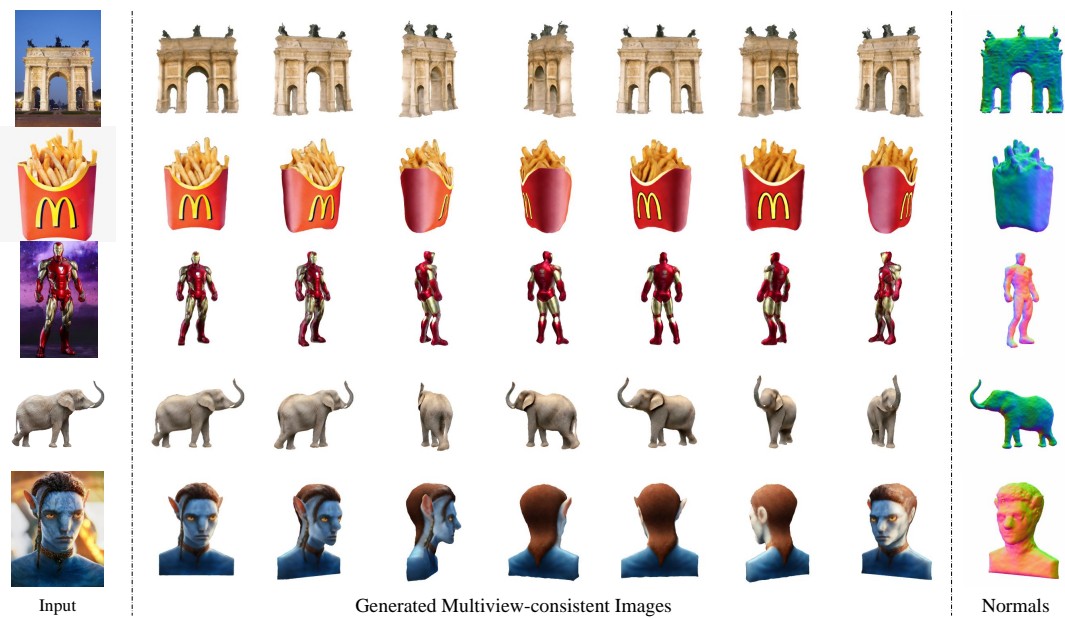

**Figure 1: The reconstructed highly consistent 3D assets from a single image of Consistent123. Rendered 3D models are presented by seven views (middle part) and normals (right part). Please visit the supplementary material for more high-quality 3D assets reconstructed by Consistent123.**

## ABSTRACT

Reconstructing 3D objects from a single image guided by pretrained diffusion models has demonstrated promising outcomes. However, due to utilizing the case-agnostic rigid strategy, their generalization ability to arbitrary cases and the 3D consistency of reconstruction are still poor. In this work, we propose Consistent123, a case-aware two-stage method for highly consistent 3D asset reconstruction from one image with both 2D and 3D diffusion priors. In the first stage, Consistent123 utilizes only 3D structural priors for sufficient geometry exploitation, with a CLIP-based case-aware adaptive detection mechanism embedded within this process. In the second stage, 2D texture priors are introduced and progressively take on a dominant guiding role, delicately sculpting the details of the 3D model. Consistent123 aligns more closely with the evolving trends in guidance requirements, adaptively providing adequate 3D geometric initialization and suitable 2D texture refinement for different objects. Consistent123 can obtain highly 3D-consistent reconstruction and exhibits strong generalization ability across various objects. Qualitative and quantitative experiments show that our method significantly outperforms state-of-the-art image-to-3D methods.

## CCS CONCEPTS

• **Computing methodologies** → **Computer vision**.

## KEYWORDS

3D reconstruction, image-to-3D, diffusion prior, case-aware optimization

## 1 INTRODUCTION

Humans possess an exceptional perceptual ability to rapidly infer the complete 3D shape and surface details of depicted objects in an image at a glance. This remarkable capability is attributed to years of visual world comprehension and the accumulation of prior knowledge. The experienced 3D artists can craft intricate 3D models from images, however, this demands hundreds of hours of manual effort. In this study, we aim to efficiently generate highly consistent 3D model from a single image. This endeavor promises to furnish a potent adjunct for 3D creation and offers a swift means of procuring 3D objects for virtual three-dimensional environment construction.

Despite decades of extensive research efforts [6, 14, 15, 18, 32], the task of reconstructing 3D structure and texture from a single viewpoint remains inherently challenging due to its ill-posed nature. To address this challenge, one category of approaches relies on costly 3D annotations obtained through CAD software or tailored domain-specific prior knowledge [33, 40], e.g. human and clothing templates, which contribute to consistent results while also limiting applicability to arbitrary objects. Another cue [12, 13, 20, 30], harnesses the generalization ability of 2D generation models like CLIP [22] and Stable Diffusion [26]. However, RealFusion [13] and Make-it-3D [30] suffer from severe multi-face issue, that is, the face appears at many views of the 3D model. With 3D structure prior, Zero-1-to-3 [12] and Magic123 [20] can stably recover the 3D structure of an object, but struggle to obtain highly consistent reconstruction. All these methods do not take into account the unique characteristics of object, and utilize fixed strategy for different cases. These case-agnostic approaches face difficulty in adapting optimization strategies to arbitrary objects.

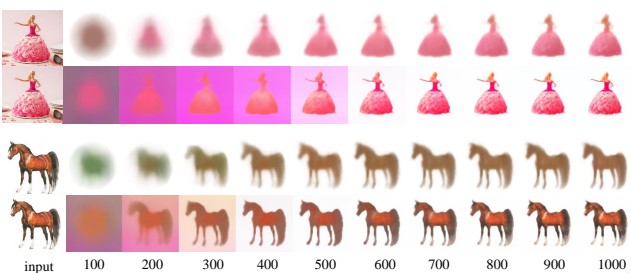

**Figure 2: The observation of optimization. For each case, the top row shows the optimization process using 2D diffusion priors, and the bottom row using 3D diffusion priors.**

Distinctly, our objective is to establish a versatile approach applicable to a broad spectrum of objects, endowed with the capability to dynamically adapt guidance strategy according to the extent of reconstruction progress. To achieve this aim, we draw attention to two pivotal **observations**: **(1)** Across various objects, a case-aware optimization phase, driven solely by 3D structural prior in the early stage, ensures the fidelity and consistency of the eventual reconstruction. **(2)** During the reconstruction process, the initial focus lies on capturing the object's overall structure, followed by the meticulous refinement of geometric shape and texture details, as illustrated in Fig 2.

Considering these, we propose *Consistent123*, a novel approach for one image to highly consistent 3D asset using case-aware 2D and 3D diffusion priors. Specifically, Consistent123 takes two stages. *Stage 1*: Consistent123 initializes the 3D content solely with 3D prior, thereby mitigating any disruption from 2D prior in structure exploitation. This process involves a case-aware boundary judgement, where we periodically sample the 3D content from fixed perspectives and measure their similarity with textual information. Once the changing rate of the similarity falls below a threshold, Consistent123 switches to stage 2. *Stage 2*: Consistent123 optimizes the 3D content with dynamic prior, namely the combination of 2D and 3D prior. Our rationale is to reduce the emphasis on 3D prior

over time, while accentuating the significance of 2D prior, which serve as the principal guidance for exploring texture intricacies. Consistent123 adaptively tailors an continuous optimization procedure for different input, facilitating the creation of exceptionally coherent 3D assets.

We evaluate Consistent123 on the RealFusion15 [13] dataset and our collected C10 dataset. Through quantitative and qualitative analysis, we demonstrate the superiority of Consistent123 when compared to state-of-the-art methods. In summary, our contributions can be summarized as follows:

- We propose a case-aware image-to-3D method, **Consistent123**, which aligns more effectively with the demands of prior knowledge. It places a heightened emphasis on 3D structural guidance in the initial stage and progressively integrates 2D texture details in the subsequent stage.
- Consistent123 incorporates an adaptive detection mechanism, eliminating the necessity for manual adjustments to the 3D-to-2D prior ratio. This mechanism autonomously identifies the conclusion of 3D optimization and seamlessly transitions to a 3D-to-2D reduction strategy, improving its applicability across objects with diverse geometric and textural characteristics.
- Consistent123 demonstrates excellent 3D consistency in contrast to purely 3D, purely 2D, and 3D-2D fusion methodologies. Furthermore, our approach yields superior geometric and textural quality.

## 2 RELATED WORK

### 2.1 Text-to-3D Generation

Generating 3D models is a challenging task, often hindered by the scarcity of 3D data. As an alternative, researchers have turned to 2D visual models, which are more readily available. One such approach is to use the CLIP model [22], which has a unique cross-modal matching mechanism that can align input text with rendered perspective images. CLIP-Mesh [16] directly employed CLIP to optimize the geometry and textures of meshes. Dream Fields [7] and CLIP-NeRF [31] utilized the neural implicit representation, NeRF [15], as the optimization target for CLIP.

Due to the promising performance of the Diffusion model in 2D image generation [23, 26, 34], some studies have extended its application to 3D generation. DreamFusion [19] directly used a 2D diffusion model to optimize the alignment between various rendered perspectives and text with SDS loss, thereby generating 3D objects that match the input text. Magic3D [11] used the two-stage optimization with diffusion model to get a higher resolution result. 3DFuse [27] generated a 2D image as a reference and introduced a 3D prior based on the generated image. It also incorporated optimization with a prompt embedding to maintain consistency across different perspectives. TEXTure [24] generated textures using a depth-to-image diffusion model and blended textures from various perspectives using a Trimap. Rodin [33] and ETRIS [38] bridged the gap between vision and language with CLIP, and achieved a unified 3D diffusion model for text-conditioned and image-conditioned 3D generation. DreamAvatar [1] transformed the observation space to a standard space with a human prior and used a diffusion model to optimize NeRF for each rendered perspective.

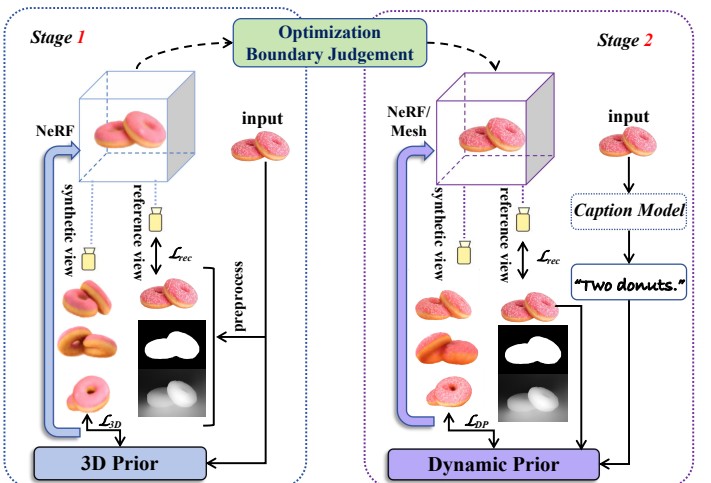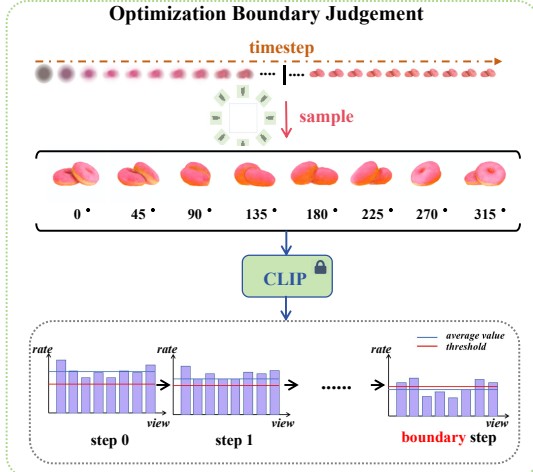

Figure 3: The framework of Consistent123. Consistent123 consists of two stages. In the first stage, we take advantage of 3D prior to optimize the geometry of 3D object. With the help of an optimization boundary judgment mechanism based on CLIP, we ensure the geometry initial optimization process is well conducted. Then, in the second stage, the output from the last stage continues to be optimized by the fusion of 2D prior and 3D prior in a specific ratio based on timestep, which is also named Dynamic Prior. To access a high-consistence and high-quality asset, we employ enhanced representation like Mesh instead of NeRF in the final period of optimization. The eventual result of the framework has correct geometry and exquisite texture from visual observation.

## 2.2 Single Image 3D Reconstruction

Single-image 3D reconstruction has been a challenging problem in the fields of graphics and computer vision, due to the scarcity of sufficient information. To address this issue, researchers have explored various approaches, including the use of 3D-aware GANs and Diffusion models. Some work [2, 36, 37, 39] leveraged 3D-aware GANs to perform 3D face generation with GAN inversion techniques [25, 35]. Other works used Diffusion models to generate new perspectives in reconstruction. Rodin [33] proposed a 3D diffusion model for high-quality 3D content creation, which is trained on synthetic 3D data. Zero-1-to-3 [12] fine-tuned Stable Diffusion with injected camera parameters on a large 3D dataset [3] to learn novel view synthesis.

Another line of work adopted 2D diffusion prior to directly optimize a 3D object without the need for large-scale 3D training data. These approaches represent promising avenues for addressing the challenge of single-image 3D reconstruction. As a seminal work, Make-It-3D [30] used an image caption model [10] to generate text descriptions of the input image. The researchers then optimized the generation of novel views with SDS loss, as well as introducing a denoised CLIP loss to maintain consistency among different views. Meanwhile, RealFusion [13] utilized textual inversion to optimize prompt embedding from input images and then employed SDS loss to optimize the generation of new perspectives. Magic123 [20] leveraged a rough 3D prior generated by Zero-1-to-3 [12] and combined it with textual inversion to optimize prompt embedding using SDS loss with fixed weighting.

## 3 METHODOLOGY

As shown in Fig 3, the optimization process of Consistent123 can be categorized from a perspective standpoint into two phases: the reference view and the novel view. In the reference viewpoint, we primarily employ the input image as the basis for reconstruction, a topic comprehensively addressed in Section 3.1. The optimization of the novel view unfolds across two distinct stages. These two stages are thoroughly explored in Sections 3.2 and Section 3.3, respectively. The resultant model output consistently exhibits a high degree of 3D consistency and exceptional texture quality.

## 3.1 Reference View Reconstruction

Imported a 2D RGB image, Consistent123 adopts a preprocess operation to get derivative ground truth which can be used in the loss calculation in the reference view. We utilize pretrained model [5, 9] to acquire the demerger $\mathbf{I}^{gt}$, the binary mask $\mathbf{M}^{gt}$ and the depth of object $\mathbf{D}^{gt}$. $\mathcal{L}_{rgb}$ ensures the similarity between the input image and the rendered reference view image. Mean Squared Error (MSE) loss is leveraged to calculate the $\mathcal{L}_{rgb}$ in the form as follows:

$$\mathcal{L}_{rgb} = \|\mathbf{I}^{gt} - \mathcal{G}_{\theta}\left(v^r\right)\|_2^2 \tag{1}$$

where $\mathcal{G}_{\theta}$ stands for the representation model in the optimization process, $v^r$ represents the viewpoint of reference view in the rendering process. The design of $\mathcal{L}_{mask}$ likewise employs MSE to operate calculation whose concrete expression as follows:

$$\mathcal{L}_{mask} = \|\mathbf{M}^{gt} - \mathbf{M}\left(\mathcal{G}_{\theta}\left(v^r\right)\right)\|_2^2 \tag{2}$$

where $\mathbf{M}\left(\cdot\right)$ means the operation of extracting the mask of the rendered image. Seeing that the method of using depth prior in the former of this area, we decide to adopt the normalized negative Pearson correlation between $\mathbf{D}^{gt}$ and the rendered depth map $d^r$ as $\mathcal{L}_{depth}$.

$$\mathcal{L}_{depth} = 1 - \frac{\text{cov}(\mathbf{M}\left(\mathbf{D}^{gt}\right), \mathbf{M}\left(d^r\right))}{\sigma(\mathbf{M}\left(\mathbf{D}^{gt}\right))\sigma(\mathbf{M}\left(d^r\right))} \tag{3}$$

where cov $(\cdot)$ denotes covariance and $\sigma(\cdot)$ measures standard deviation.

Given three vital parts of reference view reconstruction loss, we merge them into a modified form of expression:

$$\mathcal{L}_{rec} = \lambda_{rgb}\mathcal{L}_{rgb} + \lambda_{mask}\mathcal{L}_{mask} + \lambda_{depth}\mathcal{L}_{depth} \qquad (4)$$

where $\lambda_{rgb}$, $\lambda_{mask}$ and $\lambda_{depth}$ are controllable parameters which are used to regulate the ratio of each supervision. With the help of merged loss $\mathcal{L}_{rec}$, we can restore a high detail and correct geometry target on the reference viewpoint.

## 3.2 Optimization Boundary Judgement

The optimization process illustrated in Fig 2 demonstrates the efficiency of 3D structural priors in capturing the shape of object, and the 3D priors play a crucial role mainly in the initial stage of reconstruction. To ensure the comprehensive recovery of the object's shape as depicted in the image, we establish a structural initialization stage, namely stage 1, where only 3D structural priors guide the optimization. The guidance of the 3D prior can be expressed as the gradient which is used to update the parameter $\theta$:

$$\nabla_{\theta}\mathcal{L}_{3D}(\phi, \mathcal{G}_{\theta}) = \mathbb{E}_{t,\epsilon}\left[w(t)\left(\epsilon_{\phi}\left(\mathbf{z}_t; \mathbf{I}^r, t, R, T\right) - \epsilon\right)\frac{\partial \mathbf{I}}{\partial \theta}\right] \qquad (5)$$

where $t$ denotes the noise level, $\mathbf{z}_t$ is the noisy latent generated by adding random Gaussian noise to the rendered view $\mathbf{I}$, $\mathbf{I}^r$ represents the reference view, $R$ and $T$ mean the rotation and translation parameters of the camera. The function $w(t)$ corresponds to a weighting function, while $\epsilon_{\phi}$ and $\epsilon$ respectively denote the noise prediction value generated by the U-Net component of the 2D diffusion model and the ground truth noise. During stage 1, 2D priors are deliberately excluded, effectively mitigating the multi-face issue. The output of this stage is 3D content with high-quality structure, yet it significantly lags in terms of texture fidelity compared to the image representation. That's mainly because of the deficiency of texture information, which is primarily driven by 2D priors.

Consequently, we embed a case-aware CLIP-based detection mechanism within stage 1 to determine whether the shape of the current 3D content has been accurately reconstructed. If so, a transition is made to stage 2, with 2D priors introduced gradually. During the first-stage training, we conduct boundary judgement at specific iterations. Specifically, we periodically perform detection at intervals of $h$ iterations, set to 20 in our experiments. For each detection step $k$, we render the current 3D content from different viewpoints, and then calculate the average similarity score between these images and textual descriptions using the CLIP model:

$$\mathcal{S}_{CLIP}^{k}\left(y, \mathcal{G}_{\theta}^{k}\right) = \frac{1}{|V|}\sum_{v\in V}\varepsilon_{CLIP}\left(\mathcal{G}_{\theta}^{k}(v)\right)\cdot\varphi_{CLIP}(y) \qquad (6)$$

where $y$ is the description of the reference image, and $v$ is a rendering perspective belonging to sample views set $V$. $\varepsilon_{CLIP}$ is a CLIP image encoder and $\varphi_{CLIP}$ is a CLIP text encoder. To determine whether the shape of the current 3D content has been adequately recovered, we compute the moving average of changing rate of $\mathcal{S}_{CLIP}$:

$$R^{k} = \frac{1}{L}\sum_{i=k-L+1}^{k}\left(\mathcal{S}_{CLIP}^{i} - \mathcal{S}_{CLIP}^{i-1}\right)/\mathcal{S}_{CLIP}^{i-1} \qquad (7)$$

where $L$ is the size of the sliding window. When this rate falls below a threshold $\delta$, the current 3D content is considered to possess a structure similar to that represented in the image.

## 3.3 Dynamic Prior

Recognizing that 3D prior optimization is characterized by consistent structure guidance but weak texture exploration, while 2D prior optimization leads to high texture fidelity but may occasionally diverge from the input image, we posit these two priors exhibit complementarity, each benefiting the quality of the final 3D model. Consequently, in Stage 2, we introduce a 2D diffusion model as the guiding 2D prior to enrich the texture details of the 3D object. Throughout the optimization process, the 2D diffusion model primarily employs Score Distillation Sampling (SDS) [19] loss to bridge the gap between predicted noise and ground truth noise. This concept is elucidated as the follows:

$$\nabla_{\theta}\mathcal{L}_{2D}(\phi, \mathcal{G}_{\theta}) = \mathbb{E}_{t,\epsilon}\left[w(t)\left(\epsilon_{\phi}\left(\mathbf{z}_t; y, t\right) - \epsilon\right)\frac{\partial \mathbf{z}}{\partial \mathbf{I}}\frac{\partial \mathbf{I}}{\partial \theta}\right] \qquad (8)$$

where $y$, originating from either user observations or the output of a caption model, represents the text prompt describing the 3D object. However, we have observed that, in the stage 2, when the optimization relies solely on the 2D prior, the resulting 3D asset often exhibits an unfaithful appearance. This is attributed to the low-resolution output of stage 1 possessing poor low-level information such as color, shading, and texture, which makes room for 2D prior to provide high-resolution but unfaithful guidance. Moreover, the alignment relationship between the input text prompt and each individual novel view which is waiting to be optimized by the 2D prior varies. This variability leads the 2D prior to introduce certain unfaithful details, which we refer to as the 'Over Imagination' issue. Consequently, the eventual output typically maintains a reasonable structure but displays an unfaithful novel view, resulting in an inconsistent appearance.

To resolve the above problem, we incorporate 3D prior and 2D prior in an incremental trade-off method instead of only using 2D diffusion model in stage 2, which we call it **Dynamic Prior**. More specifically, we design a timestep-based dynamic integration strategy of two kinds of prior to gradually introduce exquisite guidance information while maintaining its faithfulness to input image. The loss formula of dynamic prior using both $\mathcal{L}_{3D}$ and $\mathcal{L}_{2D}$ is as follows:

$$\mathcal{L}_{DP} = e^{-\frac{t}{T}}\mathcal{L}_{3D} + \left(1 - e^{-\frac{t}{T}}\right)\mathcal{L}_{2D} \qquad (9)$$

where $T$ represents total timesteps of optimization. As shown in Equation (9), we determine the weighting coefficients of two losses using an exponential form which is dependent on the training iteration $t$. As $t$ increases, $\mathcal{L}_{3D}$ which is primarily contributing structural information undergoes a gradual reduction in weight, while $\mathcal{L}_{2D}$ which is mainly responsible for optimizing texture information exhibits a progressive increase of influence. We have also considered expressing $\mathcal{L}_{DP}$ in the form of other basis functions, but extensive experimental results have shown that the expression in Equation (9) yields many excellent and impressive results, and more details of the comparison can be found in Section 4.4. Compared to single prior or fixed ratio prior, the outputs of Consistent123 are

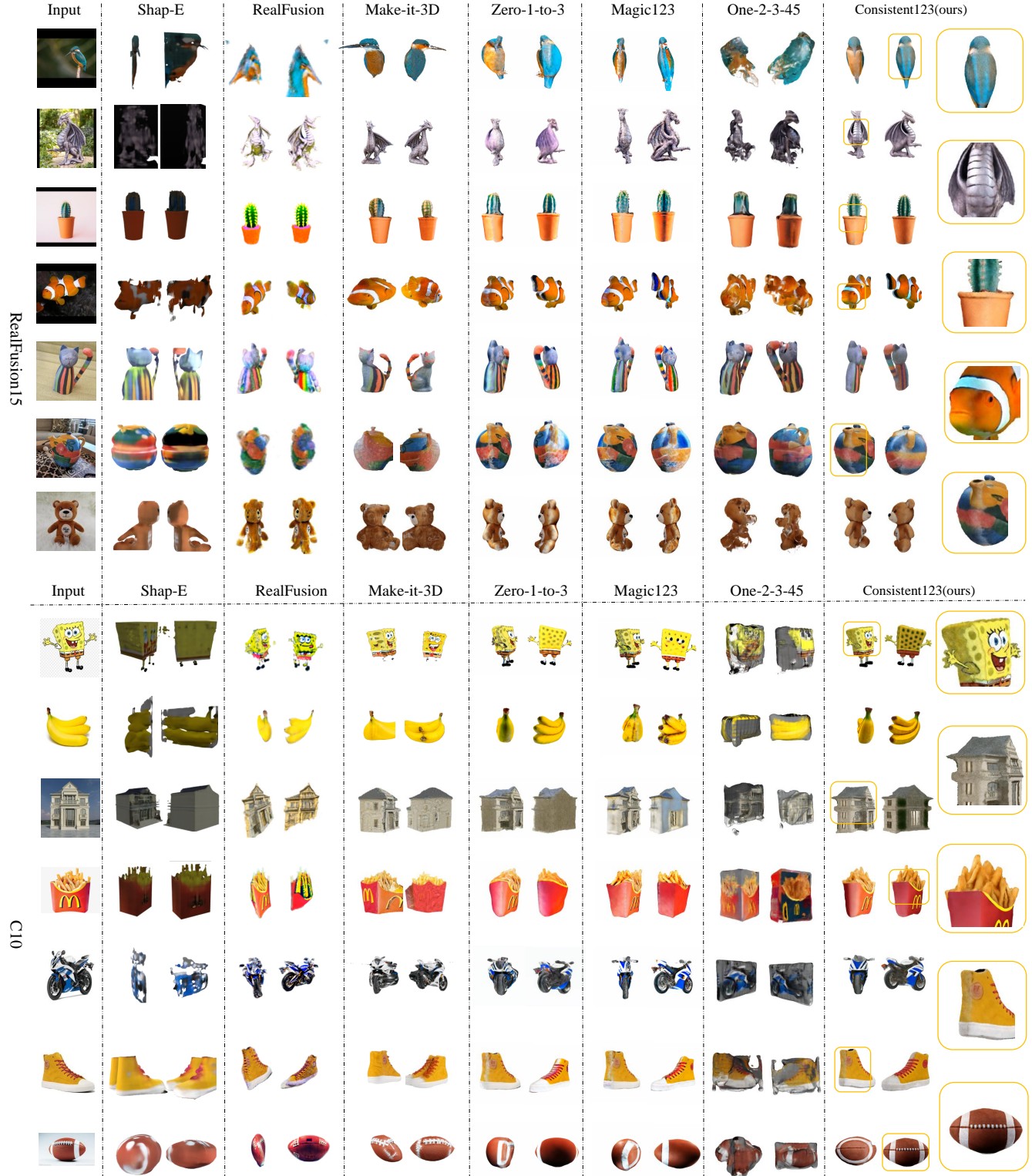

**Figure 4: Qualitative comparison vs SOTA methods. The results on the RealFusion15 dataset is shown on top, and results on the C10 dataset on the bottom. We randomly sample 2 novel views to showcase, and reference view and other views are included in the supplementary material.**

**Table 1: Quantitative results on the RealFusion15 and C10 datasets. Make-it-3D uses CLIP similarity to supervise the training, so its value$^{\dagger}$ is not considered for Make-it-3D in the comparison.**

| Dataset | Methods | Shap-E | RealFusion | Make-it-3D | Zero-1-to-3 | Magic123 | One-2-3-45 | Consistent123(ours) |
|---|---|---|---|---|---|---|---|---|
| **RealFusion15** | CLIP-Similarity↑ | 0.544 | 0.735 | $0.839^{\dagger}$ | 0.759 | 0.747 | 0.679 | **0.844** |
| | PSNR↑ | 6.749 | 20.216 | 20.010 | 25.386 | 25.637 | 13.754 | **25.682** |
| | LPIPS↓ | 0.598 | 0.197 | 0.119 | 0.068 | 0.062 | 0.329 | **0.056** |
| **C10** | CLIP-Similarity↑ | 0.508 | 0.680 | $0.824^{\dagger}$ | 0.700 | 0.751 | 0.673 | **0.770** |
| | PSNR↑ | 6.239 | 22.355 | 19.412 | 18.292 | 15.538 | 14.081 | **25.327** |
| | LPIPS↓ | 0.639 | 0.140 | 0.120 | 0.229 | 0.197 | 0.277 | **0.054** |

more consistent and exquisite from the perspective of texture and geometry.

## 4 EXPERIMENTS

### 4.1 Implementation Details

For the diffusion prior, we adopt the open-source Stable Diffusion [26] of version 2.1 as 2D prior, and employ the Zero-1-to-3 [12] as the 3D prior. We use Instant-NGP [17] to implement the NeRF representation and for mesh rendering , we utilize DMTet [28], a hybrid SDF-Mesh representation. The rendering resolutions are configured as 128 × 128 for NeRF and 1024 × 1024 for mesh. Following the camera sampling approach adopted in Dreamfusion [19], we sample the reference view with a 25% probability and the novel views with a 75% probability. The weighting coefficients $\lambda_{rgb}$, $\lambda_{mask}$, $\lambda_{depth}$ are set to 1000, 500, 10 respectively. For the case-aware detection mechanism, we sample from 8 viewpoints each time, that is $V = \{0°, 45°, 90°, 135°, 180°, 225°, 270°, 315°\}$. The sliding window size $L$ is set to 5 and the threshold $\delta$ of 0.00025. We use Adam optimizer with a learning rate of 0.001 throughout the reconstruction. For an image, the entire training process with 10,000 iterations takes approximately 30 minutes on a single NVIDIA A100 GPU.

### 4.2 Comparison with State-of-the-art

**Datasets.** We consider a classic benchmark, **RealFusion15**, released by RealFusion [13]. RealFusion15 consists of 15 images featuring a variety of subjects. In addition, we introduced a **C10** dataset consisting of 100 images collected from 10 categories which covers a wider range of items. These 10 categories broadly encompass common objects found in daily life, including fruits, balls, furniture, scenes, flora and fauna, food, transportation, clothing and footwear, cartoon characters, and artwork. Thus, the results on C10 can serve as an effective evaluation of the method's generalization ability. For 3D evaluation, we utilize **Google Scanned Objects(GSO)** [4], a 3D object dataset containing 3D meshes, to evaluate the accuracy of the reconstructed meshes. Specifically, for a 3D object, we take the officially provided front-rendered image as input to the model. The object mesh is used as 3D ground truth to conduct the evaluation.

**Baselines and metrics.** We choose 6 SOTA methods, namely Shap-E [8], RealFusion [13], Make-it-3D [30], Zero-1-to-3 [12], Magic123 [20], and One-2-3-45 [?], for extensive comparison. For Shap-E and one-2-3-45, we leverage their open-source API on the huggingface website for 3D reconstruction. We use an improved implementation [29] of Zero-1-to-3, and the original released code

for other works. On RealFusion15 and C10, we report three metrics, namely **CLIP-similarity** [21], **PSNR** and **LPIPS** [41]. CLIP-similarity quantifies the average CLIP distance between the rendered images from the reference view, and serves as a measure of 3D consistency by assessing appearance similarity across novel views and the reference view. PSNR and LPIPS assess the reconstruction quality and perceptual similarity at the reference view. On GSO, we measure **Chamfer Distance**(CD), **volumetric IoU**, **multi-view PSNR** and **multi-view LPIPS**. The ground truth 3D mesh and the reconstructed 3D mesh are first normalized within the unit cube.

**Qualitative comparison.** We present a comprehensive set of qualitative results featuring 14 images drawn from the RealFusion15 and C10 datasets in Fig 4. In contrast to our method, Shap-E and One-2-3-45 fail to recover the proper structure. RealFusion often yields flat 3D results with colors and shapes that exhibit little resemblance to the input image. Make-it-3D displays competitive texture quality but grapples with a prominent issue of multi-face. For instance, when reconstructing objects like teddy bears and Spongeboy, it introduces facial features at different novel views, which should only appear in the reference view. Zero-1-to-3 and Magic123 produce visually plausible structures, but the consistency of texture among all views, especially in side views, is poor. For example, in the cases of fish and rugby, their textures fail to achieve a smooth transition when observed from the side view. In contrast, our methodology excels in generating 3D models that not only exhibit semantic consistency with the input image but also maintain a high degree of consistency in terms of both texture and geometry across all views.

**Quantitative comparison.** As demonstrated in Table 1, on the RealFusion15 dataset, Consistent123 attains the most favorable results in the CLIP-Similarity metric which gain an increment of **11.2%** compared to the original SOTA, signifying that our method yields the most consistent 3D models. Regarding reference view reconstruction, Consistent123 performs comparably to Magic123 and Zero-1-to-3, and significantly outperforms others. On the C10 dataset, encompassing images from 10 distinct categories, Consistent123 outpaces its counterparts by a substantial margin across all evaluation metrics. Moreover, there is a notable enhancement in CLIP-Similarity, accompanied by an improvement of **2.972** in PSNR and **0.066** in LPIPS metrics when compared to the previously top-performing model, which underscore robust generalization capability of Consistent123 across diverse object categories. Turning attention to Table 2, on GSO, Consistent123 achieves the highest CD and IoU, demonstrating our structure closely aligns with the 3D ground truth. Notably, Consistent123 excels in both multi-view

**Table 2: Quantitative results on the GSO dataset.**

| Methods | Zero-1-to-3 | Magic123 | One-2-3-45 | ours |
|---------|-------------|----------|------------|------|
| CD↓ | 0.0959 | 0.0989 | 0.0427 | **0.0416** |
| IoU↑ | 0.3364 | 0.3189 | 0.4941 | **0.5020** |
| PSNR↑ | 17.110 | 17.363 | 17.408 | **17.684** |
| LPIPS↓ | 0.289 | 0.285 | 0.303 | **0.278** |

PSNR and LPIPS, which signifies the texture quality of our 3D outcomes surpasses that of other methods.

## 4.3 Ablation Study of Two Stage Optimization

In this section, we explore the significance of boundary judgment mechanism. We divide the reconstruction process into three parts, namely: 3D structural initialization, boundary judgment, and dynamic prior-based optimization. In cases where boundary judgment is absent, the optimization process can be categorized into two approaches: full 3D structural initialization (boundary at the training starting point) or full dynamic prior-based optimization (boundary at the training endpoint), denoted as Consistent123$_{3D}$ and Consistent123$_{dynamic}$ respectively. As illustrated in Fig 5, without the guidance of 2D texture priors, Consistent123$_{3D}$ produces visually unrealistic colors in the new view of the car, and in the absence of 3D structural initialization, Consistent123$_{dynamic}$ exhibits inconsistency and multi-face issue in Mona Lisa's face. In contrast, results with boundary judgment showcase superiority in both texture and structure.

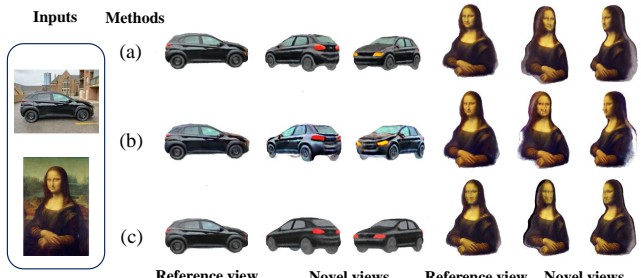

**Figure 5: The ablation of two stages. (a) Consistent123, (b) Consistent123$_{3D}$ and (c) Consistent123$_{dynamic}$.**

## 4.4 Ablation Study of Dynamic Prior

Dynamic prior refers to the method of dynamically adjusting the ratio of 2D and 3D priors based on different time steps during the optimization process. Depending on the transformation method, we compare the optimization effects of three different approaches: exponential (Equation (9)), linear(Equation (10)) and logarithmic (Equation (11)). We assessed them across ten categories, each comprising 5 images from the RealFusion15 and C10 datasets. As shown in the Table 3, the exponential variation process, which is the our adopted method, can achieve a higher CLIP-Similarity on most of the categories, which to some extent reflects the reconstruction consistency. The actual reconstruction results also support this, as the exponential variation method can effectively mitigate the

multi-head problem, leading to higher reconstruction quality and better consistency.

$$\mathcal{L}_{linear} = \frac{t}{T}\mathcal{L}_{3D} + \left(1 - \frac{t}{T}\right)\mathcal{L}_{2D} \tag{10}$$

$$\mathcal{L}_{log} = \log_2 \frac{t}{T}\, \mathcal{L}_{3D} + \left(1 - \log_2 \frac{t}{T}\right)\mathcal{L}_{2D} \tag{11}$$

The key difference between exponential transformation and the other two lies in the fact that exponential transformation can inject 2D priors more quickly. The focus of dynamic priors is to optimize the quality and consistency of the reconstruction with 2D texture priors while maintaining the correctness of the 3D structure. In stage 2, we have observed that establishing the dominant role of 2D priors early on facilitates the extraction of texture details. Subsequently, gradually increasing the weights of 2D priors slowly ensures minimal texture distortion, thereby ensuring the reconstruction of high-quality textures. The exponential weight variation process seamlessly aligns with this objective.

## 4.5 Ablation Study of Hyperparameters

We ablate the weighting coefficients $\lambda_{rgb}, \lambda_{mask}, \lambda_{depth}$ in the reference view reconstruction loss equation on the RealFusion15 and C10 datasets. We set up 5 parameter settings: (a){$\lambda_{rgb} = 1000, \lambda_{mask} = 500, \lambda_{depth} = 0$}, (b){$\lambda_{rgb} = 10, \lambda_{mask} = 1000, \lambda_{depth} = 500$}, (c){$\lambda_{rgb} = 500, \lambda_{mask} = 10, \lambda_{depth} = 1000$}, (d){$\lambda_{rgb} = 800, \lambda_{mask} = 600, \lambda_{depth} = 400$}, (default){$\lambda_{rgb} = 1000, \lambda_{mask} = 500, \lambda_{depth} = 10$}. The quantitative results of different parameter settings are shown in Table 4.

In addition, for boundary judgment, we ablate the threshold $\delta$ and the sliding window size $L$. We set up 5 settings: (a){$L = 1, \delta = 0.000025$}, (b){$L = 20, \delta = 0.000025$}, (c){$L = 5, \delta = 0.0001$}, (d){$L = 5, \delta = 0.001$}, (default){$L = 5, \delta = 0.00025$}. The quantitative results of different parameter settings are shown in Table 5.

**More interesting exploration.** In this part, some interesting experiments are conducted to further demonstrate the effectiveness of our two-stage optimization method with boundary judgment and dynamic prior. Specifically, we devise three different optimization strategies: (a) remove boundary judgments and perform stage transition in a specific iteration step (3000 in our experiment), (b) the weight of the 2D prior is always set to 1.0 in stage 2 (dynamic prior stage), (c) the two stages are combined into a single stage and the 3D object is optimized using the following dynamic prior.

$$\mathcal{L}_{DP} = e^{-\frac{t}{\sigma T}}\mathcal{L}_{3D} + \left(1 - e^{-\frac{t}{\sigma T}}\right)\mathcal{L}_{2D} \tag{12}$$

where $\sigma$ takes the value of 1.5, which represents the initial phase of optimization with a higher weighted 3D prior. We show the results of these strategies on the RealFusion15 and C10 datasets in Table 6. Our default strategy achieves the best results on all metrics. The results of strategy (a) prove the necessity of our boundary judgment mechanism, and it is not appropriate to use the same optimization process for all cases. The results of strategy (b) show that the gradual introduction of 2D prior is necessary. Strategy (c) leverages 2D prior from the beginning, leading to worse results, which characterize the necessity of our 3D initialization phase.

**Table 3: Ablation Study of Dynamic Prior on the RealFusion15 and C10 datasets.**

| Methods | Metrics/class | ball | biont | furniture | cartoon | fruit | statue | food | vehicle | costume | scene | average |
|---|---|---|---|---|---|---|---|---|---|---|---|---|
| **log** | CLIP-Similarity↑ | 0.79 | 0.85 | **0.58** | 0.77 | 0.87 | 0.71 | 0.87 | 0.74 | **0.67** | 0.68 | 0.76 |
| | PSNR↑ | 26.45 | 25.46 | 23.19 | 23.97 | 24.62 | 22.94 | 27.33 | 24.24 | **26.14** | 21.71 | 24.59 |
| | LPIPS↓ | **0.04** | 0.06 | **0.12** | **0.06** | 0.06 | 0.11 | **0.03** | 0.07 | 0.06 | 0.10 | 0.07 |
| **linear** | CLIP-Similarity↑ | 0.82 | 0.85 | 0.55 | 0.74 | **0.88** | 0.73 | **0.88** | 0.72 | 0.65 | 0.70 | 0.76 |
| | PSNR↑ | 26.32 | 25.51 | 22.96 | 23.43 | 25.31 | **25.71** | 27.41 | 24.57 | 25.36 | 21.63 | 24.96 |
| | LPIPS↓ | **0.04** | 0.05 | 0.13 | 0.09 | **0.04** | **0.06** | **0.03** | 0.07 | 0.06 | 0.10 | 0.07 |
| **exp** | CLIP-Similarity↑ | **0.87** | **0.88** | 0.54 | **0.78** | 0.87 | **0.77** | **0.88** | **0.76** | **0.67** | **0.72** | **0.79** |
| | PSNR↑ | **27.50** | **26.09** | **23.28** | **24.29** | 25.39 | 25.63 | 27.02 | **25.16** | 25.65 | 21.78 | **25.30** |
| | LPIPS↓ | **0.04** | **0.04** | **0.12** | **0.06** | 0.05 | 0.07 | 0.04 | **0.05** | **0.05** | **0.09** | **0.06** |

**Table 4: Quantitative ablation about $\lambda_{rgb}$, $\lambda_{mask}$, and $\lambda_{depth}$ on the RealFusion15 and C10 datasets.**

| Dataset | setting | (a) | (b) | (c) | (d) | default |
|---|---|---|---|---|---|---|
| **RealFusion15** | CLIP-Similarity↑ | 0.812 | 0.763 | 0.701 | 0.795 | **0.844** |
| | PSNR↑ | 25.097 | 18.597 | 16.961 | 23.288 | **25.682** |
| | LPIPS↓ | 0.069 | 0.174 | 0.206 | 0.085 | **0.056** |
| **C10** | CLIP-Similarity↑ | 0.748 | 0.669 | 0.649 | 0.724 | **0.770** |
| | PSNR↑ | 25.177 | 16.857 | 17.129 | 22.196 | **25.327** |
| | LPIPS↓ | 0.067 | 0.217 | 0.216 | 0.098 | **0.054** |

**Table 5: Quantitative ablation about $L$ and $\delta$ on the RealFusion15 and C10 datasets.**

| Dataset | setting | (a) | (b) | (c) | (d) | default |
|---|---|---|---|---|---|---|
| **RealFusion15** | CLIP-Similarity↑ | 0.812 | 0.814 | 0.817 | 0.815 | **0.844** |
| | PSNR↑ | 25.464 | 25.272 | 25.528 | 25.607 | **25.682** |
| | LPIPS↓ | 0.068 | 0.070 | 0.069 | 0.068 | **0.056** |
| **C10** | CLIP-Similarity↑ | 0.748 | 0.753 | 0.747 | 0.753 | **0.770** |
| | PSNR↑ | 25.651 | 25.151 | 25.506 | 23.398 | **25.327** |
| | LPIPS↓ | 0.065 | 0.065 | 0.060 | 0.066 | **0.054** |

**Table 6: Quantitative ablation about optimization strategies on the RealFusion15 and C10 datasets.**

| Dataset | setting | (a) | (b) | (c) | default |
|---|---|---|---|---|---|
| **RealFusion15** | CLIP-Similarity↑ | 0.813 | 0.813 | 0.825 | **0.844** |
| | PSNR↑ | 25.604 | 24.836 | 24.812 | **25.682** |
| | LPIPS↓ | 0.068 | 0.070 | 0.069 | **0.056** |
| **C10** | CLIP-Similarity↑ | 0.746 | 0.730 | 0.765 | **0.770** |
| | PSNR↑ | 25.066 | 25.562 | 25.253 | **25.327** |
| | LPIPS↓ | 0.063 | 0.067 | 0.062 | **0.054** |

## 4.6 User Study

Due to the absence of ground-truth 3D models, we conducted a perceptual study to compare Consistent123 against SOTA baselines. we conducted a user study comprising 784 feedbacks from 56 users to gather statistical data. From the RealFusion15 and C10 datasets, we carefully selected 14 representative cases to gauge user preferences. Participants were tasked with selecting the best result that represents the texture and structure of the object depicted in the image. To quantify the likelihood of participants favoring SOTA methods over Consistent123, we present the corresponding results

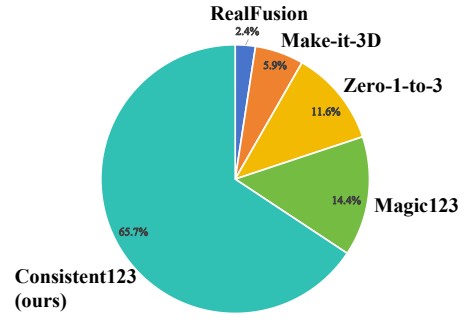

**Figure 6: User Study. The collected results of preference.**

in Fig 6. Our method demonstrates superior performance compared to the alternatives, exhibiting a **65.7%** advantage in the user study.

## 5 CONCLUSION AND DISCUSSION

**Conclusion**. In this study, we introduce Consistent123, a two-stage framework designed for achieving highly detailed and consistent 3D reconstructions from single images. By recognizing the complementary nature of 3D and 2D priors during the optimization process, we have devised a training trade-off strategy that prioritizes initial geometry optimization with 3D priors, followed by the gradual incorporation of exquisite guidance from 2D priors over the course of optimization. Between the two optimization stages, we employ a large-scale pretrained image-text pair model as a discriminator for multi-view samples to ensure that the 3D object gains sufficient geometry guidance before undergoing dynamic prior optimization in stage 2. The formulation of our dynamic prior is determined through the exploration of various foundational function forms, with a subsequent comparison of their categorized experimental results. Our approach demonstrates enhanced 3D consistency, encompassing both structural and textural aspects, as demonstrated on existing benchmark datasets and those we have curated.

**Limitation**. Our study reveals two key limitations. Firstly, during stage 1, heavy reliance on 3D priors influences the 3D object, with reconstruction quality notably affected by the input image's viewpoint. Secondly, output quality depends on the description of asset in stage 2. Finer-grained descriptions enhance output consistency, while overly brief or ambiguous descriptions lead to the 'Over Imagination' issue in Stable Diffusion [26], introducing inaccurate details.

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
