# OpenReview forum: "Consistent123: One Image to Highly Consistent 3D Asset Using Case-Aware Diffusion Priors"
_acmmm.org/ACMMM/2024/Conference — MM2024 Poster_

### Official Review · Reviewer_gp5n · 2024-04-29

**Rating:** 4
**Confidence:** 3

**Summary:**

This paper focuses on the task of 3D reconstruction from a single image. It aims to keep 3D reconstruction consistency with the help of 2D and 3D prior knowledge. Existing 3D reconstruction approaches either rely on CAD models or ignore the category information of objects, leading to poor generalization ability on unseen objects. This paper proposes a new method with multiple stages to use the power of prior knowledge for obtaining well-reconstructed objects of high quality.

**Strengths:**

1.	The proposed new approach is reasonable and is designed elaborately.
2.	The proposed new approach can achieve the best performance on two challenging datasets.
3.	This paper is well-organized and comprehensive, including detailed experimental results and an analysis of limitations.

**Limitations:**

1.	While the authors claim that the 3D priors can guide on structural and the 2D priors can provide the texture details, it is hard to be convinced of this since in Figure 2, in which case the bottom rows with 3D priors in both two cases showing better structural and texture results compared to the top rows with 2D priors.
2.	The above problem also leads to this one: please first clearly explain the difference between 2D and 3D priors, and testify the claimed difference between them, then start to introduce the proposed approach which contains three stages. This problem is located at the paragraph starting from the Line #162. More evidence should be given to prove that 2D and 3D priors are experts in different perspectives.
3.	The GPT-based polishers are not prohibited, but using it too much will cause the text to be hard to read since too many unusual words are used and sometimes the content of descriptive words is not appropriate, which may exaggerate the contribution of some small designs.
4.	Minor: The abbreviations should be consistent. For example, it should be Fig. 1 and Tab.1, not Table 1. Besides, it should be Fig. 1, not Fig 1.

**Suitability:**

3

---

### Official Review · Reviewer_2y1r · 2024-05-19

**Rating:** 5
**Confidence:** 3

**Summary:**

Consistent123 introduces an innovative framework capable of generating highly consistent 3D models from a single image. This method employs a two-stage optimization process that integrates 2D and 3D diffusion priors, enhanced by a case-aware optimization strategy to improve the accuracy and consistency of the reconstruction. The authors conducted qualitative and quantitative experiments, demonstrating that Consistent123 significantly outperforms state-of-the-art image-to-3D methods across various datasets. The method was evaluated using metrics such as CLIP similarity, PSNR, and LPIPS, showcasing superior performance in terms of 3D consistency and texture quality.

**Strengths:**

1. Consistent123 introduces a novel two-stage framework that leverages both 2D and 3D diffusion priors, providing a fresh perspective on the image-to-3D conversion challenge.
2. The method's superiority is supported by rigorous quantitative metrics and qualitative analysis, showcasing its effectiveness against existing techniques.
3. Positive user feedback from perceptual studies indicates that Consistent123 meets user expectations for 3D reconstruction quality.
4. Well-written manuscript.

**Limitations:**

1. There is a need to provide additional details, for example, in section 3.3, the acquisition of the text prompt $y$ is not clearly explained, and $y$ is necessary for SDS.
2. What is the significance of introducing text-to-3D in the "Related Work" section? Since Consistent123 uses SDS, why isn't there more introduction to SDS in section 2.1?
3. Some details need to be refined, for example, there is a citation issue with One-2-3-45 on line 634.

**Suitability:**

2

---

### Official Review · Reviewer_medg · 2024-05-25

**Rating:** 5
**Confidence:** 3

**Summary:**

This paper proposes a 2-stages optimizing strategy to reconstruct 3D objects from single image.

**Strengths:**

1. The peper tells a good story. First it provide the observition of differences between reconstructing using 3D and 2D prior, then proposes the 2-stage optimizing strategy and stage-trasfering boundary which is simple but useful;
2. Provides comprehensive implementation details;
3. Provides rich experiments, including user studies;

**Limitations:**

The results in the Ablation Study did not fully demonstrate the advantages of introducing 2D priors in 2-stage compared to the pure 3D strategy. We hope that 2D priors can not only correct colors , but also enrich details and textures.

**Suitability:**

3

---

### Meta-Review · Area_Chair_tah9 · 2024-06-29

**Recommendation:** Accept (Poster)
**Confidence:** 4

**Metareview:**

A 2-stage optimization strategy is proposed to reconstruct a consistent 3D object form a single 2D image.
Pros: both 2D and 3D diffusion priors are engaged. Comprehensive empirical evaluations are conducted, including positive user study results. The proposed method shows good performance across two benchmark datasets. The write up is easy to follow.
Cons: more and detailed ablation study. Better clarification of the differences between the 2D and the 3D priors.